# Incidence and Survival Outcomes of Colorectal Cancer in Long-Term Metformin Users with Diabetes: A Population-Based Cohort Study Using a Common Data Model

**DOI:** 10.3390/jpm12040584

**Published:** 2022-04-05

**Authors:** Seung In Seo, Tae Jun Kim, Chan Hyuk Park, Chang Seok Bang, Kyung Joo Lee, Jinseob Kim, Hyon Hee Kim, Woon Geon Shin

**Affiliations:** 1Division of Gastroenterology, Department of Internal Medicine, Kangdong Sacred Heart Hospital, Hallym University College of Medicine, 445, Gil-dong, Kangdong-gu, Seoul 05355, Korea; doctorssi@kdh.or.kr (S.I.S.); lee30553969@gmail.com (K.J.L.); 2Institute for Liver and Digestive Diseases, Hallym University, Chuncheon 24252, Korea; cloudslove@hanmail.net; 3Department of Internal Medicine, Samsung Medical Center, Sungkyunkwan University School of Medicine, Seoul 06351, Korea; taejunk91@gmail.com; 4Department of Internal Medicine, Hanyang University Guri Hospital, Hanyang University College of Medicine, Guri 11923, Korea; yesable7@gmail.com; 5Department of Internal Medicine, Chuncheon Sacred Heart Hospital, Hallym University College of Medicine, Chuncheon 24253, Korea; 6Department of Epidemiology, School of Public Health, Seoul National University, Seoul 03080, Korea; jinseob2kim@gmail.com; 7Department of Statistics and Information Science, Dongduk Women’s University, Seoul 02748, Korea; hyonheekim@gmail.com

**Keywords:** colorectal cancer, metformin, chemoprevention

## Abstract

Background and aims: Previous studies have reported that metformin use in patients with diabetes mellitus may reduce the risk of colorectal cancer (CRC) incidence and prognosis; however, the evidence is not definite. This population-based cohort study aimed to investigate whether metformin reduces the risk of CRC incidence and prognosis in patients with diabetes mellitus using a common data model of the Korean National Health Insurance Service database from 2002 to 2013. Methods: Patients who used metformin for at least 6 months were defined as metformin users. The primary outcome was CRC incidence, and the secondary outcomes were the all-cause and CRC-specific mortality. Cox proportional hazard model was performed and large-scaled propensity score matching was used to control for potential confounding factors. Results: During the follow-up period of 81,738 person-years, the incidence rates (per 1000 person-years) of CRC were 5.18 and 8.12 in metformin users and non-users, respectively (*p* = 0.001). In the propensity score matched cohort, the risk of CRC incidence in metformin users was significantly lower than in non-users (hazard ratio (HR), 0.58; 95% CI (confidence interval), 0.47–0.71). In the sensitivity analysis, the lag period extending to 1 year showed similar results (HR: 0.63, 95% CI: 0.51–0.79). The all-cause mortality was significantly lower in metformin users than in non-users (HR: 0.71, 95% CI: 0.64–0.78); CRC-related mortality was also lower among metformin users. However, there was no significant difference (HR: 0.55, 95% CI: 0.26–1.08). Conclusions: Metformin use was associated with a reduced risk of CRC incidence and improved overall survival.

## 1. Introduction

Despite frequent screening for colorectal cancer (CRC) and population-level lifestyle modifications such as declines in smoking, CRC is the third most common malignancy and the second leading cause of cancer-related deaths worldwide [1]. Fecal immunochemical test-based screening achieved a 10% reduction in CRC incidence and screening colonoscopy showed a 74% reduction in distal CRC and a 27% reduction in proximal colon cancer incidence [2,3]. Although the screening tools are effective for CRC prevention and mortality reduction, there is an unmet need to identify effective chemopreventive agents against CRC as a primary prevention method.

Several chemopreventive medications against CRC have been studied. Among them, the most randomized controlled trials and commonly used medications are aspirin, nonsteroidal anti-inflammatory drugs, statins, and metformin. A pooled analysis of five trials found that low-dose aspirin reduces long-term CRC incidence and mortality [4]. However, aspirin use was associated with serious bleeding events such as cerebral or gastrointestinal hemorrhage. In a meta-analysis, aspirin was associated with an increased risk of cerebral (odds ratio (OR), 1.34; 95% confidence interval (CI) 1.07–1.70) and gastrointestinal hemorrhage (OR, 1.59; 95% CI, 1.32–1.91) [5]. Numerous epidemiologic studies have reported an association of statins with the decreased risk of various malignancies; however, there are conflicting results for CRC [6,7,8,9]. A recent meta-analysis of observational studies reported that metformin use in patients with type 2 diabetes mellitus (DM) was associated with a reduced incidence of CRC [10]. Another meta-analysis of observational studies suggested that metformin use may improve cancer-specific survival and overall survival [11]. However, the observational studies in the meta-analysis are susceptible to confounding by various co-medications and comorbidities; therefore, residual confounding effects exist. Hence, we aimed to investigate the protective effect of metformin on the CRC incidence, overall survival, and CRC-specific survival in patients with DM, using a nationwide database, while ensuring all measures were taken to avoid potential bias including a new-drug user model, negative controls, and large-scale propensity score matching.

## 2. Methods

### 2.1. Study Design and Database

We conducted a population-based cohort study using a national cohort database in South Korea known as the National Health Insurance Service–National Sample Cohort, which was converted to the Observational Medical Outcomes Partnership Common Data Model (OMOP-CDM). The national sample cohort includes 1,025,340 participants, comprising 2.2% of the total eligible Korean population in 2002, and followed for 11 years until 2013. This included their demographic profile, health insurance claims data, death registry, disability registry, and national health check-up data [12,13]. The NHIS–NSC data was converted into the OMOP-CDM model (denominated National Health Insurance Service-Common Data Model (NHIS-CDM)), and the database was validated in previous studies [14,15,16,17]. The protocol of the current study was approved by the Institutional Review Board of Kangdong Sacred Heart Hospital (IRB no. 2019-05-014).

### 2.2. Cohort Definition

The target cohort involved patients with DM who were metformin users for over 6 months. Continuous drug exposures were achieved by allowing less than 30-day gaps between prescriptions. All patients were diagnosed with DM before 1 year of drug prescription.

The index date was defined as the first date of each drug prescription. The exclusion criteria are as follows: (1) history of any malignant neoplasm within 3 years before cohort entry; (2) an observation period of fewer than 365 days before cohort entry; and (3) age < 18 years at cohort entry. The comparative cohort was defined as the use of other anti-diabetic drugs including insulin and oral hypoglycemic agent (OHA) for over 6 months except metformin before 30 days of cohort entry in patients with DM. The OHA includes sulfonylurea, meglitinide, thiazolidinedione, DPP4-inhibitor, and alpha-glucosidase inhibitor. The exclusion criteria were the same as that for the target cohort.

The target and comparative cohort were censored if the observation periods ended in the database or if the patients were diagnosed with CRC. The comparative cohort was censored if they took metformin additionally. We performed subgroup analysis in patients with metformin use for over 1 year to identify the cumulative dose-response relationship.

## 3. Outcomes

The outcome cohort was defined as patients who were newly diagnosed with CRC at least after 6 months from the index date. CRC was identified using the ICD-10 diagnosis codes C18.0–C20. The primary outcome was CRC incidence, and the secondary outcomes were all-cause and CRC-specific mortality.

## 4. Main Statistical Analysis

Large-scale propensity score (PS) matching was performed with logistic regression models with the L1 penalty hyperparameter, selected through 10-fold cross-validation using high-performance computing [18]. The included covariates were as follows: age group, sex, index year, Charlson comorbidity index, all recorded drugs before 1 year of cohort entry, all recorded diagnostic before 1 year of cohort entry.

Cox proportional hazard model was used to calculate the hazard ratio (HRs) with 95% confidence intervals (CIs). The Kaplan–Meier method was used to estimate the cumulative incidence rates, and the cumulative incidence between two groups was compared using the log-rank test. Two-sided *p*-values < 0.05 were considered statistically significant in all comparisons. All analyses were performed using ATLAS ver. 2.7.2 and R statistical software (version 3.6.1 for Windows; R Foundation for Statistical Computing, Berlin, Germany).

### Sensitivity Analyses

Sensitivity analysis was conducted to assess the robustness of the results, including propensity-score matching with different lag periods and propensity score stratification. We performed empirical calibration of the *p*-values by fitting an empirical null distribution to point estimates of the negative control outcomes, which were assumed not to be associated with the target or comparative cohorts. A total of 90 selected negative control outcomes are listed in Appendix A.

## 5. Results

### 5.1. Patients Characteristics

After large-scale PS matching (metformin group = 43,367, non-metformin group = 20,801), 8201 patients were grouped in the target and comparative cohorts for the analysis of CRC incidence (Figure 1).

The patients’ characteristics and Standard mean difference (SMD) before and after PS matching are shown in Table 1. A total of 12,532 covariates were included in the PS matching, and the maximum SMD after PS adjustment was 0.06 (Appendix A). The metformin use and non-use groups were well balanced after large-scale PS matching. The most common medical history before cohort entry was a hypertensive disorder, followed by acute respiratory disease and hyperlipidemia (Table 1). The most common medications used were antibiotics, drugs for acid-related disorders, and antithrombotic agents (Table 1).

The Charlson comorbidity index was 4.31 and 4.26 in the metformin and non-metformin groups, respectively. The patients’ characteristics of overall and CRC-related mortality were not different compared with CRC incidence. The distribution of HRs in negative control outcomes are shown in Appendix A. Nearly all negative control outcomes were not significantly different between the two groups.

### 5.2. Incidence of CRC between Metformin Users and Non-Users

The overall results are summarized in Table 2. During a median follow-up of 4.9 years, the incidence of CRC among metformin users was significantly lower than among non-users (CRC incidence; metformin vs. non-metformin, 220/42,458 person-years vs. 319/39,280 person-years, HR: 0.58, 95% CI: 0.47–0.71, *p* = 0.001). The incidence of CRC among metformin users for over 1 year was significantly lower than among non-users (HR: 0.72, 95% CI: 0.58–0.88, *p* = 0.001) (Table 2).

However, it showed no cumulative dose response relationship. The cumulative incidence of CRC is shown in Figure 2A.

### 5.3. All-Cause and CRC-Related Mortality between Metformin Users and Non-Users

In the analysis of mortality, all 8264 patients regardless of metformin use were included in the final analysis after PS matching. During a median follow-up of 5.1 years, the all-cause mortality was significantly lower in metformin users than in non-users (all-cause mortality; metformin vs. non-metformin, 1111/43,254 person-years vs. 1449/40,567 person-years, HR: 0.71, 95% CI: 0.64–0.78, *p* = 0.001) (Table 2) (Figure 2B). The all-cause mortality rate in metformin users over 1 year was also significantly lower than in non-users (HR: 0.70, 95% CI: 0.62–0.78, *p* = 0.001) (Table 2).

The CRC-related mortality rate was also lower in metformin-users, although there was no significant difference (CRC-related mortality; metformin vs. non-metformin, 22/43,254 person-years vs. 37/40,567 person-years, HR: 0.55, 95% CI: 0.26–1.08, *p* = 0.09) (Table 2) (Figure 2C). However, our database did not include TNM data of the neoplasm at onset and the type of treatment patients received.

### 5.4. Sensitivity Analyses of Metformin Users for Over 6 Months

In addition to the main analysis, we applied the lag period extending to 1 year and PS stratification to ensure the robustness of our results. The results of sensitivity analysis were described in Table 3.

For the CRC incidence, the lag period extending to 1 year (HR: 0.63, 95% CI: 0.51–0.79) and PS stratification (6 months-lag period, HR: 0.65, 95% CI: 0.57–0.75; 1 year-lag period, HR: 0.68, 95% CI: 0.59–0.79) both showed lower incidence rates in metformin-users (Table 3).

For the all-cause mortality, the lag period extending to 1 year (HR: 0.72, 95% CI: 0.66–0.80) and PS stratification (6 months-lag period, HR: 0.68, 95% CI: 0.64–0.73; 1 year-lag period, HR: 0.68, 95% CI: 0.64–0.73) both showed lower all-cause mortality rates in metformin-users (Table 3).

For the CRC-related mortality rate, only the results of PS stratification with a 6 month-lag period showed a statistical significance (HR: 0.60, 95% CI: 0.39–0.93) (Table 3).

## 6. Discussion

In this propensity score matched cohort study using nationwide database, metformin use in patients with DM led to a 42% reduction in CRC incidence compared with that of the non-users. After expanding the lag period to 1 year, metformin use was significantly associated with a lower risk of CRC. In addition, metformin use was significantly associated with improved overall survival compared with patients with diabetes who did not use metformin.

Previous studies have examined the chemopreventive role of metformin in CRC and have provided inconsistent results [10,19,20,21,22]. In a randomized controlled study on the cancer incidence of metformin in comparison with sulfonylureas and rosiglitazone, metformin use caused no difference in cancer incidence compared with the use of rosiglitazone and glyburide in patients with DM [20]. Another randomized controlled study on the effect of metformin in colon polyp recurrence was performed. In contrast to the former context, metformin use for 1 year by patients without DM reduced the risk of colon polyp recurrence after polypectomy [19]. In a retrospective nationwide cohort study in Taiwan, metformin use for ≥3 years was associated with a reduced risk of CRC incidence compared with non-users [21]. Another large-scale retrospective cohort study in the U.K. reported that metformin users had a similar risk of CRC incidence compared with that of other antidiabetic users [22]. These inconsistent results may be due to heterogenous and often incomplete control of confounding factors and limited sample size of studies. Indeed, patients with DM are a heterogenous population with various co-medication and comorbidities. In addition, observational studies are susceptible to selection bias between metformin users and non-users. Our metformin users and non-users group were well matched after extensive propensity score matching, including all diagnoses, medications, and comorbidity index before cohort entry. Therefore, we could substantially eliminate the bias and limitations of confounders.

Regarding the effects of metformin on CRC-related survival, a number of observational studies have investigated the association between metformin use and CRC-specific survival and overall survival. Metformin users with type 2 DM and CRC had a 30% improvement in overall survival compared with that of patients with diabetes, as well as when compared with that of users of other diabetic drugs [23]. In a Korean cohort study, metformin use in patients with CRC and DM is associated with improved CRC-specific and overall survival [24]. In contrast, a population-based cohort study of 1197 CRC patients with type 2 DM demonstrated that metformin use was not associated with cancer-specific survival [25]. A recent meta-analysis of 17 observational studies reported a protective association between metformin use and CRC-specific mortality. In our study, the CRC-specific mortality was lower in metformin users; however, there was no significant difference. Further randomized controlled studies are needed to verify the association between metformin use and CRC-related survival.

There are several plausible mechanisms for the chemopreventive effect of metformin against CRC. Metformin can suppress CRC cell growth through inhibition of Myc protein levels and protein synthesis by metformin-mediated reduction of MAP kinase-interacting protein kinase 1 [26]. Metformin also inhibits CRC cell proliferation via metformin-activated AMPK, which is capable of exerting a number of different effects on β-catenin signaling [27,28]. One study suggested that metformin attenuates CRC stemness and epithelial-mesenchymal transition by inhibiting the Wnt3a/β-catenin pathway [29]. Another study proposed an anticancer mechanism for metformin wherein cancer cells proliferate by upregulating glucose uptake and increasing the rate of glycolytic activity. Conversely, metformin causes mechanistic reduction in glycolytic rate, which may potentially serve anticancer purposes [30].

Our study has several strengths. First, it was designed to overcome various biases. We utilized a new user design with 1 year of observation period before cohort entry to reduce immortal time bias [31]. Further, we defined the comparative group as patients taking other anti-diabetic drugs except for metformin in DM to avoid indication bias. We applied the same design to the analyses of all-cause and CRC-related mortality. Second, we showed a variety of sensitivity analyses with different lag periods and analytic methods to overcome the limitations of an observational study. The findings were consistent with the main results. Third, our study was based on a national database converted to OMOP-CDM. Therefore, the study can be extended to other databases with common analytic R code [32]. Lastly, using the large-scaled propensity score matching, we could adjust for various covariates, which could influence the incidence of CRC or prognosis. The number of covariates was 12,532, and the maximum SMD after propensity score matching was 0.06. This implies that we could strengthen the comparability between metformin users and non-users with large-scaled propensity score matching.

Despite the strengths, there are potential limitations. First, we could not adjust the laboratory findings such as hemoglobin A1c or glucose level. Hence, the severity of DM was not reflected in the study results. Nevertheless, we included all recorded comorbidities, therefore, the end organ damage of DM, such as retinopathy or nephropathy, was adjusted as covariates. In addition, the Charlson comorbidity index includes the complication of DM. Thus, the severity of DM was adjusted in some degree. Second, even though we made efforts to avoid biases, the observational study may have had residual biases. Third, there was lack of information on the tumor histology, stage, and colonoscopic findings in the database. Lastly, we could not consider the dose of each medicine or time of prescription, which might lead to the biased results.

In conclusion, this nationwide study showed that metformin use was associated with a significantly lower risk of CRC incidence and improved overall survival in patients with DM. Metformin might be considered in patients with DM to prevent CRC and reduce all-cause mortality in patients with DM. However, further randomized controlled trials are warranted to confirm the chemopreventive effects of metformin on CRC.

## Figures and Tables

**Figure 1 jpm-12-00584-f001:**
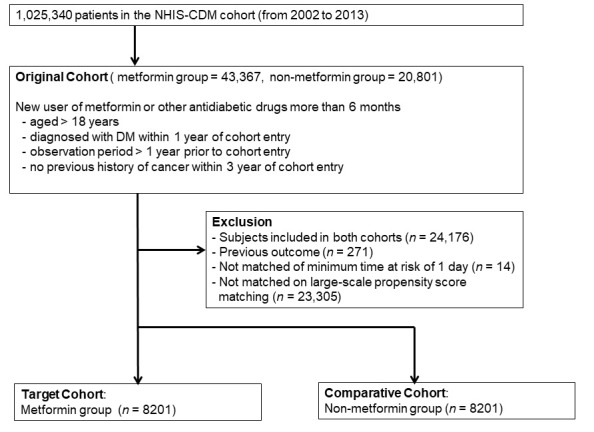
Flow chart in the analysis of colorectal cancer incidence. NHIS-CDM, National Health Insurance Service-Common Data Model; DM, diabetes mellitus.

**Figure 2 jpm-12-00584-f002:**
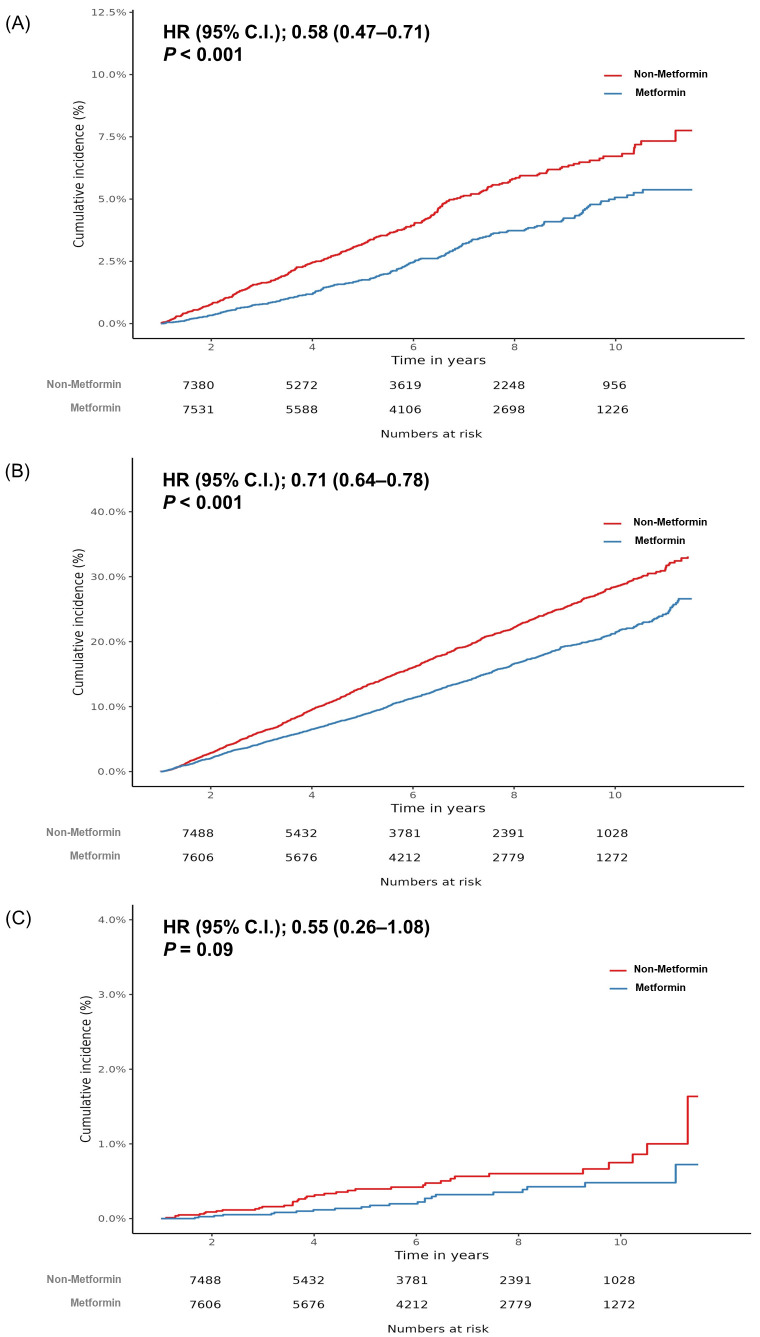
Kaplan–Meier curve for the cumulative incidence of colorectal cancer (**A**), all-cause mortality (**B**), and colorectal cancer-related mortality (**C**) between metformin users and non-users. HR, hazard ratio; CI, confidence interval.

**Table 1 jpm-12-00584-t001:** Baseline characteristics between metformin users and non-users in the analysis of CRC incidence.

Characteristic	Before PS Adjustment			After PS Adjustment		
	Metformin (*n* = 31,071)	Non-Metformin (*n* = 8636)	SMD	Metformin (*n* = 8201)	Non-Metformin *(n* = 8201)	SMD
Age group (years, %)						
45–49	11	8.2	0.09	8.5	8.6	0
50–54	14.8	10.4	0.13	10.4	10.6	−0.01
55–59	14.6	12	0.08	12.5	12.3	0.01
60–64	14.7	15.2	−0.01	14.8	15.1	−0.01
65–69	13.1	16	−0.08	15.7	15.9	−0.01
70–74	9.7	13.4	−0.11	13.4	13.3	0
75–79	5.8	9.7	−0.15	9.5	9.3	0.01
Sex: female, %	46.6	48.4	−0.04	49.1	48.5	0.01
Cigarette smoker, %	6.3	5.2	0.05	5.2	5.3	−0.01
Alcoholics, %	12.9	10.2	0.08	9.6	10.6	−0.03
Medical history, %						
Acute respiratory disease	56	54.3	0.03	54	54.4	−0.01
Chronic liver disease	8.8	9.7	−0.03	9.7	9.5	0.01
Renal impairment	1.5	5.2	−0.21	2.6	2.3	0.02
Chronic kidney disease	0.5	4.0	−0.23	1.5	1.2	0.02
Depressive disorder	7.5	9	−0.05	8.3	8.5	−0.01
Gastroesophageal reflux disease	8.5	7.9	0.02	7.6	7.8	−0.01
Hyperlipidemia	50.1	43.1	0.14	41.2	42.5	−0.03
Hypertensive disorder	55.9	63.9	−0.16	63.5	62.7	0.02
Osteoarthritis	15.1	17.2	−0.06	17.4	17.2	0
Visual system disorder	36	37.2	−0.02	36.9	36.5	0.01
Retinal disorder	6.7	7.2	−0.01	7.1	6.7	0.01
Cerebrovascular disease	5.4	6.9	−0.06	6.6	6.5	0
Heart disease	22.8	29.5	−0.15	28.5	27.7	0.02
Heart failure	5.1	8.2	−0.12	7.5	7.1	0.02
Ischemic heart disease	12.6	17.6	−0.14	16.8	16.2	0.02
Peripheral vascular disease	14.4	15	−0.02	14.8	14.6	0
Obesity	0.3	0.1	0.03	0.2	0.1	0.02
Medication use, %						
Agents acting on the renin-angiotensin system	35.8	38.1	−0.05	36.9	36.7	0
Antibacterials for systemic use	65	65.7	−0.01	66.2	64.9	0.03
Antidepressants	12	13.3	−0.04	12.9	12.8	0
Antiepileptics	7.7	9.1	−0.05	8.4	8.5	0
Anti-inflammatory and antirheumatic products	55.5	57.9	−0.05	58.3	57.6	0.01
Antithrombotic agents	60.6	61.7	−0.02	61.2	60.7	0.01
Beta blocking agents	20.4	26.7	−0.15	25.7	24.9	0.02
Calcium channel blockers	35.2	44.2	−0.18	43.9	42.5	0.03
Diuretics	31.6	36.8	−0.11	36.3	34.8	0.03
Drugs for acid-related disorders	60.9	62.5	−0.03	62.6	61.4	0.02
Drugs for obstructive airway diseases	37.2	39.1	−0.04	38	38.7	−0.01
Drugs used in diabetes	11.5	5.6	0.21	5.6	5.7	−0.01
Lipid-modifying agents	36.1	30.7	0.12	28.6	30.2	−0.04
Opioids	41	41.3	0	41.1	40.6	0.01
Psycholeptics	43.8	48.6	−0.1	47.8	47.6	0
Charlson comorbidity index–Romano adaptation	4.23	4.41	−0.07	4.31	4.26	0.02

Values are presented as proportion of the patients (%). Abbreviations: CRC, colorectal cancer; PS, propensity score; SMD, standard mean difference.

**Table 2 jpm-12-00584-t002:** Incidence of colorectal cancer, all-cause mortality, and colorectal cancer-related mortality in metformin users.

	No. of Participants	Person-Years	No. of Case	Incidence Rate ^a^	HR	95% CI	*p*-Value
CRC incidence							
Metformin ≥ 6 months	8201	42,458	220	5.18	0.58	0.47–0.71	0.001
Non-metformin	8201	39,280	319	8.12	Ref.		
Metformin ≥ 1 year	8021	42,073	223	5.30	0.72	0.58–0.88	0.001
Non-metformin	8021	39,444	291	7.38	Ref.		
All-cause mortality							
Metformin ≥ 6 months	8264	43,254	1111	25.69	0.71	0.64–0.78	0.001
Non-metformin	8264	40,567	1449	35.72	Ref.		
Metformin ≥ 1 year	8077	42,791	919	21.48	0.70	0.62–0.78	0.001
Non-metformin	8077	40,575	1192	29.38	Ref.		
CRC-related mortality							
Metformin≥ 6 months	8264	43,254	22	0.51	0.55	0.26–1.08	0.09
Non-metformin	8264	40,567	37	0.91	Ref.		
Metformin ≥ 1 year	8077	42,791	19	0.44	0.85	0.37–1.89	0.69
Non-metformin	8077	40,575	32	0.79	Ref.		

^a^ Incidence rate expressed per 1000 person-years. Abbreviations: CRC, colorectal cancer; CI, confidence interval; HR, hazard ratio.

**Table 3 jpm-12-00584-t003:** Sensitivity analyses of metformin users for over 6 months.

Analysis	Lag Period	HR	95% CI	*p*-Value
CRC incidence				
PS matching (main analysis)	6 months	0.58	0.47–0.71	0.001
PS matching	1 year	0.63	0.51–0.79	0.001
PS stratification	6 months	0.65	0.57–0.75	0.001
PS stratification	1 year	0.68	0.59–0.79	0.001
All-cause mortality				
PS matching (main analysis)	6 months	0.71	0.64–0.78	0.001
PS matching	1 year	0.72	0.66–0.80	0.001
PS stratification	6 months	0.68	0.64–0.73	0.001
PS stratification	1 year	0.68	0.64–0.73	0.001
CRC-related mortality				
PS matching (main analysis)	6 months	0.55	0.26–1.08	0.09
PS matching	1 year	0.65	0.32–1.29	0.23
PS stratification	6 months	0.60	0.39–0.93	0.02
PS stratification	1 year	0.67	0.43–1.06	0.08

Abbreviations: CRC, colorectal cancer; CI, confidence interval; HR, hazard ratio.

## Data Availability

Data are available upon reasonable request.

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
