# Peer review of "Incidence and Survival Outcomes of Colorectal Cancer in Long-Term Metformin Users with Diabetes: A Population-Based Cohort Study Using a Common Data Model"

_jpm, 2022, doi:10.3390/jpm12040584_

Round 1

Reviewer 1 Report

This is an observational study that examined the effect of metformin on colorectal cancer prevention using a health insurance database in South Korea. By comparing metformin-treated and non-treated groups of approximately 8,000 individuals, the study showed that metformin treatment reduced the risk of developing colorectal cancer (hazard ratio 0.58). Total mortality was also at lower risk with metformin (HR 0.71). Although the study was based on a large database and the methodology of the study itself is valid, several concerns can be raised. The first is the importance of the study. The preventive effect of metformin on colorectal cancer has already been demonstrated by numerous reports and multiple meta-analyses to date. What is needed in the future is to demonstrate the preventive effect of metformin on colorectal cancer through intervention studies rather than large-scale observational studies. This study, although relatively large, is an observational study and does not add anything new to past findings or overcome weaknesses of previous studies. Next, the authors claim to have devised a way to minimize bias by propensity score matching. However, the covariates used were age, gender, index year, comorbidity index, prescription medications, and diagnosis. For these, propensity score matching did not differ between the metformin group and the other groups. However, other factors, such as obesity (BMI), diabetes severity, and organ damage, were different between the two groups, and it is possible that these factors, but not metformin, may have influenced the risk of colorectal cancer occurrence. These points need to be discussed appropriately. In the conclusion, the statement recommending the use of metformin is an oversimplification.

Author Response

Thank you for reviewing our manuscript and inviting revision. We are resubmitting our manuscript after making a new version according to your recommendations. Revised contents were highlighted in Yellow in the text in addition to last revision.

Reviewer 1

This is an observational study that examined the effect of metformin on colorectal cancer prevention using a health insurance database in South Korea. By comparing metformin-treated and non-treated groups of approximately 8,000 individuals, the study showed that metformin treatment reduced the risk of developing colorectal cancer (hazard ratio 0.58). Total mortality was also at lower risk with metformin (HR 0.71). Although the study was based on a large database and the methodology of the study itself is valid, several concerns can be raised.

The first is the importance of the study. The preventive effect of metformin on colorectal cancer has already been demonstrated by numerous reports and multiple meta-analyses to date. What is needed in the future is to demonstrate the preventive effect of metformin on colorectal cancer through intervention studies rather than large-scale observational studies. This study, although relatively large, is an observational study and does not add anything new to past findings or overcome weaknesses of previous studies.

Answer) We absolutely agree with your opinion. In fact, causal relation between metformin and reduction of CRC incidence can be revealed through RCTs. However, these RCTs is very difficult and somewhat less feasible to conduct because of many subjects, long follow-up periods, and huge cost. The previous reports were case-control or small cohort studies having various biases such as selection, immortal-time, indication, protopathic, and/or confounding bias. Thus, we constructed new-user model and applicated extensive propensity score matching with covariates of about 12000, lag periods, and negative controls. In this study, SMDs after PS matching of less than 0.06 reflected the well-matched target and comparative cohort. Moreover, we conducted sensitive analyses with changing taking durations of metformin and lag periods (6month and 1 year) and PS stratification to confirm robustness for our results. We tried the various tools to overcome the limitations, please consider our efforts.   

Next, the authors claim to have devised a way to minimize bias by propensity score matching. However, the covariates used were age, gender, index year, comorbidity index, prescription medications, and diagnosis. For these, propensity score matching did not differ between the metformin group and the other groups. However, other factors, such as obesity (BMI), diabetes severity, and organ damage, were different between the two groups, and it is possible that these factors, but not metformin, may have influenced the risk of colorectal cancer occurrence. These points need to be discussed appropriately. In the conclusion, the statement recommending the use of metformin is an oversimplification.

Answer) Thank you very much for your valuable comments. We agree with your opinion. There was possibility more severe DM patents were included in comparative cohort, though we performed PS matching with more than 12,000 variables. However, we adjusted retinal disorder, chronic kidney disease, and obesity. We cannot adjust BMI because NHIS data have not this information, but we also matched the various medications and diseases related obesity such as hyperlipidemia or hypertension. We attached revised Table 1 as below.

Table 1. Baseline characteristics between metformin users and non-users in the analysis of CRC incidence.

Characteristic

Before PS adjustment

After PS adjustment

Metformin

(n=31,071)

Non-metformin

(n=8,636)

SMD

Metformin

(n=8,201)

Non-metformin

(n=8,201)

SMD

Cigarette smoker, %

6.3

5.2

0.05

5.2

5.3

-0.01

Alcoholics, %

12.9

10.2

0.08

9.6

10.6

-0.03

Medical history,%

Acute respiratory disease

56

54.3

0.03

54

54.4

-0.01

Chronic liver disease

8.8

9.7

-0.03

9.7

9.5

0.01

Renal impairment

1.5

5.2

-0.21

2.6

2.3

0.02

Chronic kidney disease

0.5

4.0

-0.23

1.5

1.2

0.02

Depressive disorder

7.5

9

-0.05

8.3

8.5

-0.01

Gastroesophageal reflux disease

8.5

7.9

0.02

7.6

7.8

-0.01

Hyperlipidemia

50.1

43.1

0.14

41.2

42.5

-0.03

Hypertensive disorder

55.9

63.9

-0.16

63.5

62.7

0.02

Osteoarthritis

15.1

17.2

-0.06

17.4

17.2

0

Visual system disorder

36

37.2

-0.02

36.9

36.5

0.01

Retinal disorder

6.7

7.2

-0.01

7.1

6.7

0.01

Cerebrovascular disease

5.4

6.9

-0.06

6.6

6.5

0

Heart disease

22.8

29.5

-0.15

28.5

27.7

0.02

Heart failure

5.1

8.2

-0.12

7.5

7.1

0.02

Ischemic heart disease

12.6

17.6

-0.14

16.8

16.2

0.02

Peripheral vascular disease

14.4

15

-0.02

14.8

14.6

0

Obesity

0.3

0.1

0.03

0.2

0.1

0.02

We also revised limitation of DISCUSSION as below (P9, line 262-265).

Despite the strengths, there are potential limitations. First, we could not adjust the laboratory findings such as hemoglobin A1c or glucose level. Hence, the severity of DM was not reflected in the study results. Nevertheless, we included all recorded comorbidities, therefore, the end organ damage of DM, such as retinopathy or nephropathy, was adjusted as covariates. In addition, the Charlson comorbidity index includes the complication of DM. Thus, the severity of DM was adjusted in some degree

We revised our conclusion as your suggestion (P9 line 272-274).

In conclusion, this nationwide study showed that metformin use was associated with a significantly lower risk of CRC incidence and improved overall survival in patients with DM. Metformin might be considered in patients with DM to prevent CRC and reduce mortality in patients with DM. However, further randomized controlled trials are warranted to confirm the chemopreventive effects of metformin on CRC.

Reviewer 2 Report

I thank the authors for the opportunity to evaluate their paper related to a topic that arouses a vivid interest in the literature, and it was an opportunity to study the latest publications released and in particular the meta-analyses produced in the last 5 years on the topic.

Diabetes (DM) itself is a negative prognostic factor in terms of OS in CRC . It is not feasible to perform an analysis of the patient with DM without taking into account the severity of the disease (this data cannot be objectified in the study). In clinical practice usually patients taking either hypoglycemic drugs other than metformin or insulin have more severe DM than those using metformin alone. Thus, it can be inferred that the two populations examined differ in DM severity at the outset.

The study obtains data on patients with DM from the National health Insurance service database from 2002. Patients were then followed through 2013.

With regard to the incidence

CRC was identified using the ICD-10 diagnoses codes.

P2 line 91: what is meant by "observation"? Were patients screened for colorectal neoplasms during this period? Taking into account the natural history of CRC, it is very likely that neoplasms diagnosed within the first year were already present before patients were enrolled in the study.

Do all patients take the same dose of metformin?

Table1. Even after PSM: >64% of patients take antibiotics; >55% take anti-inflammatory and antirheumatic drugs; >60% take antithrombotic drugs; >60% take antacids; >28% use lipid-modifying agents. All the listed drugs have been evaluated in many studies for their carcinogenic and protective potential.  In the paper is provided the data on how many patients take these drugs, but it is not known neither the dosage nor the duration of therapy taken.

Table 1 does not provide information on cigarette smoking behavior or daily alcohol consumption.

P3 line 97-98  How long was the follow-up of the patients?

P5 line 149-154 Whereas, one could refute the conclusion reached by explaining the difference in incidence by a greater severity of DM in patients not using metformin.

With regard to the CRC-Related motrality

The study does not provide data regarding the stage of colorectal neoplasia (TNM) at its onset, nor do we know the treatment that was offered to the patient: endoscopic polypectomy, surgery, chemotherapy.

P6 line 162 – 167 Whereas, one could refute the conclusion reached by explaining the difference in incidence by a greater severity of DM in patients not using metformin.

P8 line 215 -218  the PSM does not truly eliminate confounding factors because, as noted earlier, it fundamentally examines two populations with a disease (DM) at a different stage and then groups patients by the type of medicine they take without regard to either dose or time of use or the stage of the disease for which it is taken.

In conclusion, despite the efforts made to build a well-structured statistical analysis, the assumptions from which this part does not allow to adequately support the results obtained.

Author Response

Thank you for reviewing our manuscript and inviting revision. We are resubmitting our manuscript after making a new version according to your recommendations. Revised contents were highlighted in Yellow in the text in addition to last revision.

Reviewer 2

I thank the authors for the opportunity to evaluate their paper related to a topic that arouses a vivid interest in the literature, and it was an opportunity to study the latest publications released and in particular the meta-analyses produced in the last 5 years on the topic.

Diabetes (DM) itself is a negative prognostic factor in terms of OS in CRC . It is not feasible to perform an analysis of the patient with DM without taking into account the severity of the disease (this data cannot be objectified in the study). In clinical practice usually patients taking either hypoglycemic drugs other than metformin or insulin have more severe DM than those using metformin alone. Thus, it can be inferred that the two populations examined differ in DM severity at the outset.

The study obtains data on patients with DM from the National health Insurance service database from 2002. Patients were then followed through 2013.

Answer) Thank you very much for your valuable comments. We understand your concerns, and we revised Table 1 including the end organ damage of DM, such as retinopathy and nephropathy to adjust severity of DM.

  1. With regard to the incidence

CRC was identified using the ICD-10 diagnoses codes.

1) P2 line 91: what is meant by "observation"? Were patients screened for colorectal neoplasms during this period? Taking into account the natural history of CRC, it is very likely that neoplasms diagnosed within the first year were already present before patients were enrolled in the study.

Answer) We applicated new-user model to minimize immortal-time bias. The observation period of 1 year before cohort entry means that there is no record prescribing metformin and diagnosing CRC. Outcome cohort of CRC defined as the patients who diagnosed CRC newly at least after lag periods of 6month or 1year from index date (the first prescribing date).

2) Do all patients take the same dose of metformin?

Table1. Even after PSM: >64% of patients take antibiotics; >55% take anti-inflammatory and antirheumatic drugs; >60% take antithrombotic drugs; >60% take antacids; >28% use lipid-modifying agents. All the listed drugs have been evaluated in many studies for their carcinogenic and protective potential.  In the paper is provided the data on how many patients take these drugs, but it is not known neither the dosage nor the duration of therapy taken.

Answer) Thank you very much for your comments. As your comment, it is ideal to analyze considering the dose of antidiabetic drugs. However, it was very difficult, because we extracted drug users through ingredient. Further, we aimed to perform large scale propensity matching, and the loss of cases are inevitable to compare between well-balanced target and comparative group. Instead, we set the duration of drug use to over 180 days in target and comparative groups. We added some phrases in the discussion section as follows (P9 line 268-269);

Lastly, we could not consider the dose of each medicine or time of prescription, it might lead to the biased results.

3) Table 1 does not provide information on cigarette smoking behavior or daily alcohol consumption.

Answer) We missed this data, and we added alcohol and smoking data in Table 1.

Cigarette smoker, %

6.3

5.2

0.05

5.2

5.3

-0.01

Alcoholics, %

12.9

10.2

0.08

9.6

10.6

-0.03

4) P3 line 97-98 How long was the follow-up of the patients?

Answer) The follow-up period was added in P3 line 97-98 (median value of 4.9 years) and P6 line 164-165 (5.1 years).

5) P5 line 149-154 Whereas, one could refute the conclusion reached by explaining the difference in incidence by a greater severity of DM in patients not using metformin.

Answer) Thank you very much for your valuable comments. We agree with your opinion. There is possibility more severe DM patents were included in the comparative group. To show the proportion of end organ damage, we added covariates of retinal disorder, renal impairment, and chronic kidney disease. The proportion was different before PS matching, but it was well-matched after PS adjustment. Please check our revised Table.

Table 1. Baseline characteristics between metformin users and non-users in the analysis of CRC incidence.

Characteristic

Before PS adjustment

After PS adjustment

Metformin

(n=31,071)

Non-metformin

(n=8,636)

SMD

Metformin

(n=8,201)

Non-metformin

(n=8,201)

SMD

Medical history,%

Acute respiratory disease

56

54.3

0.03

54

54.4

-0.01

Chronic liver disease

8.8

9.7

-0.03

9.7

9.5

0.01

Renal impairment

1.5

5.2

-0.21

2.6

2.3

0.02

Chronic kidney disease

0.5

4.0

-0.23

1.5

1.2

0.02

Depressive disorder

7.5

9

-0.05

8.3

8.5

-0.01

Gastroesophageal reflux disease

8.5

7.9

0.02

7.6

7.8

-0.01

Hyperlipidemia

50.1

43.1

0.14

41.2

42.5

-0.03

Hypertensive disorder

55.9

63.9

-0.16

63.5

62.7

0.02

Osteoarthritis

15.1

17.2

-0.06

17.4

17.2

0

Visual system disorder

36

37.2

-0.02

36.9

36.5

0.01

Retinal disorder

6.7

7.2

-0.01

7.1

6.7

0.01

Cerebrovascular disease

5.4

6.9

-0.06

6.6

6.5

0

Heart disease

22.8

29.5

-0.15

28.5

27.7

0.02

Heart failure

5.1

8.2

-0.12

7.5

7.1

0.02

Ischemic heart disease

12.6

17.6

-0.14

16.8

16.2

0.02

Peripheral vascular disease

14.4

15

-0.02

14.8

14.6

0

Obesity

0.3

0.1

0.03

0.2

0.1

0.02

We also revised limitation of DISCUSSION as below.

Despite the strengths, there are potential limitations. First, we could not adjust the laboratory findings such as hemoglobin A1c or glucose level. Hence, the severity of DM was not reflected in the study results. Nevertheless, we included all recorded comorbidities, therefore, the end organ damage of DM, such as retinopathy or nephropathy, was adjusted as covariates. In addition, the Charlson comorbidity index includes the complication of DM. Thus, the severity of DM was adjusted in some degree

  1. With regard to the CRC-Related motrality

1) The study does not provide data regarding the stage of colorectal neoplasia (TNM) at its onset, nor do we know the treatment that was offered to the patient: endoscopic polypectomy, surgery, chemotherapy.

Answer) Thank you for your comment. Unfortunately, the TNM stage is not included in the NHIS-CDM database, and the endoscopic procedures are not included. We mentioned the limitation in the DISCUSSION as below. Third, there was lack of information on the tumor histology, stage, and colonoscopic findings in the database (P9 line 267-268)

2) P6 line 162 – 167 Whereas, one could refute the conclusion reached by explaining the difference in incidence by a greater severity of DM in patients not using metformin.

Answer) Thank you very much for your valuable comments. We answered in the question with regard to incidence as shown above. Please check our answer.

3) P8 line 215 -218 the PSM does not truly eliminate confounding factors because, as noted earlier, it fundamentally examines two populations with a disease (DM) at a different stage and then groups patients by the type of medicine they take without regard to either dose or time of use or the stage of the disease for which it is taken. In conclusion, despite the efforts made to build a well-structured statistical analysis, the assumptions from which this part does not allow to adequately support the results obtained.

Answer) We added your comments as limitation in DISCUSSION.

Lastly, we could not consider the dose of each medicine or time of prescription, it might lead to the biased results.

Round 2

Reviewer 2 Report

I read with pleasure the new version of the work and the comments of the authors.

I believe that the authors have  exhaustively integrated the paper.

I think it is correct to add another sentence either in P6 after line 175, or in P8 that would reiterate the lack of TNM data of the neoplasm at onset and the type of treatment the patients received.

Other clarification in Conclusions P9 line 273 "all-cause mortality," not "mortality"

Author Response

Answer) Thank you for your contribution to our manuscript. We revised the manuscript as your suggestion.

We added the sentence you suggested in P6 after line 175 as below.

However, our database did not include TNM data of the neoplasm at onset and the type of treatment patients received.

Also, the sentence was revised at P9 line 273 as below.

Metformin might be considered in patients with DM to prevent CRC and reduce all-cause mortality in patients with DM.